# Myelin Disruption, Neuroinflammation, and Oxidative Stress Induced by Sulfite in the Striatum of Rats Are Mitigated by the pan-PPAR agonist Bezafibrate

**DOI:** 10.3390/cells12121557

**Published:** 2023-06-06

**Authors:** Nícolas Manzke Glänzel, Belisa Parmeggiani, Mateus Grings, Bianca Seminotti, Morgana Brondani, Larissa D. Bobermin, César A. J. Ribeiro, André Quincozes-Santos, Jerry Vockley, Guilhian Leipnitz

**Affiliations:** 1Programa de Pós-Graduação em Ciências Biológicas: Bioquímica, Instituto de Ciências Básicas da Saúde, Universidade Federal do Rio Grande do Sul, Rua Ramiro Barcelos, 2600-Anexo, Porto Alegre 90035-003, RS, Brazil; nicolas.glanzel@gmail.com (N.M.G.); belisasp@gmail.com (B.P.); mateus.grings@gmail.com (M.G.); seminotti.bianca@gmail.com (B.S.); morganabrondani@gmail.com (M.B.); andrequincozes@ufrgs.br (A.Q.-S.); 2Division of Genetic and Genomic Medicine, Department of Pediatrics, University of Pittsburgh, Pittsburgh, PA 15261, USA; vockleyg@upmc.edu; 3Programa de Pós-Graduação Neurociências, Instituto de Ciências Básicas da Saúde, Universidade Federal do Rio Grande do Sul, Rua Ramiro Barcelos, 2600-Anexo, Porto Alegre 90035-003, RS, Brazil; larissabobermin@gmail.com; 4Natural and Humanities Sciences Center, Universidade Federal do ABC, São Bernardo do Campo 09606-070, SP, Brazil; cesar.ribeiro@ufabc.edu.br; 5Departamento de Bioquímica, Instituto de Ciências Básicas da Saúde, Universidade Federal do Rio Grande do Sul, Rua Ramiro Barcelos, 2600-Anexo, Porto Alegre 90035-003, RS, Brazil; 6Department of Human Genetics, Graduate School of Public Health, University of Pittsburgh, Pittsburgh, PA 15261, USA

**Keywords:** sulfite, myelin, neuroinflammation, oxidative stress, bezafibrate, striatum

## Abstract

Sulfite predominantly accumulates in the brain of patients with isolated sulfite oxidase (ISOD) and molybdenum cofactor (MoCD) deficiencies. Patients present with severe neurological symptoms and basal ganglia alterations, the pathophysiology of which is not fully established. Therapies are ineffective. To elucidate the pathomechanisms of ISOD and MoCD, we investigated the effects of intrastriatal administration of sulfite on myelin structure, neuroinflammation, and oxidative stress in rat striatum. Sulfite administration decreased Fluoromyelin^TM^ and myelin basic protein staining, suggesting myelin abnormalities. Sulfite also increased the staining of NG2, a protein marker of oligodendrocyte progenitor cells. In line with this, sulfite also reduced the viability of MO3.13 cells, which express oligodendroglial markers. Furthermore, sulfite altered the expression of interleukin-1β (IL-1β), interleukin-6 (IL-6), interleukin-10 (IL-10), cyclooxygenase-2 (COX-2), inducible nitric oxide synthase (iNOS) and heme oxygenase-1 (HO-1), indicating neuroinflammation and redox homeostasis disturbances. Iba1 staining, another marker of neuroinflammation, was also increased by sulfite. These data suggest that myelin changes and neuroinflammation induced by sulfite contribute to the pathophysiology of ISOD and MoCD. Notably, post-treatment with bezafibrate (BEZ), a pan-PPAR agonist, mitigated alterations in myelin markers and Iba1 staining, and IL-1β, IL-6, iNOS and HO-1 expression in the striatum. MO3.13 cell viability decrease was further prevented. Moreover, pre-treatment with BEZ also attenuated some effects. These findings show the modulation of PPAR as a potential opportunity for therapeutic intervention in these disorders.

## 1. Introduction

Isolated sulfite oxidase (SO) deficiency (ISOD) and molybdenum cofactor deficiency (MoCD) are inherited metabolic disorders caused by mutations in the gene *Suox* which encodes the SO apoenzyme (ISOD), and defects in the pathway that synthesize the SO molybdenum cofactor (MoCo), respectively. SO is a mitochondrial enzyme that oxidizes sulfite and is dependent on the molybdenum cofactor along with a heme-containing domain to transfer the electrons from sulfite to the terminal electron acceptor cytochrome *c*. Both ISOD and MoCD are biochemically characterized by the accumulation of sulfite, thiosulfate and cysteine-S-sulfate in tissues of patients, including the central nervous system. Individuals affected by MoCD further have increased levels of xanthine and hypoxanthine accompanied by low serum concentration of uric acid since they have a deficiency in the Mo-containing xanthine dehydrogenase [1,2].

Individuals with ISOD and MoCD manifest severe neurological symptoms, including seizures, axial hypotonia and/or peripheral hypertonia, feeding difficulties and psychomotor retardation, which typically leads to early death [1,2,3]. MRI abnormalities include severe encephalopathy, basal ganglia and cerebral cortex atrophy, and neuronal death. There is also evidence of demyelination in the white matter, diffuse spongiosis and gliosis in different brain regions [1,2,3,4]. Of note, white matter changes were revealed in ~90% of patients with ISOD and 60–70% with MoCD [1,5].

Sulfite accumulation has been shown to mediate the neuropathophysiology of ISOD and MoCD. Studies showed that sulfite generates high levels of reactive oxygen species, provokes lipid peroxidation, and disturbs the antioxidant system in rat brain and plasma [6,7,8,9]. Sulfite also impairs mitochondrial respiration by inhibiting glutamate and malate dehydrogenases. These effects lead to dissipation of mitochondrial membrane potential, disturbed Ca^2+^ homeostasis, decrease in NAD(P)H, and induction of apoptosis in the brain of rodents [10,11]. Additionally, intracerebral administration of sulfite disrupted mitochondrial biogenesis and MAPK pathways and induced gliosis and neuronal damage in the striatum of rats [12,13]. Mitochondrial energetic failure, elevated superoxide levels, and disturbances of mitochondrial dynamics have been demonstrated in fibroblast cell culture with *MOCS1* deficiency and SO-deficient mice and rats [12,14,15].

Current therapies for ISOD and MoCD are limited and include the use of a diet low in sulfur containing amino acids while treating symptoms non-specifically, especially seizures and spasticity. Some patients with MoCD type A respond to treatment with cyclic pyranopterin monophosphate (cPMP), an intermediate of MoCo biosynthetic pathway [16]. However, treatments for patients with other MoCD types and ISOD are lacking. 

Bezafibrate (BEZ) is a pan-peroxisome proliferator-activated receptor (pan-PPAR) agonist that may modulate the expression of genes that regulate several processes, such as antioxidant defenses, energy metabolism, mitochondrial biogenesis, inflammation, lipid homeostasis, and cellular differentiation and proliferation [17,18]. BEZ also augments the expression of peroxisome proliferator-activated receptor-gamma coactivator-1α (PGC-1α), the major regulator of mitochondrial biogenesis [19]. Owing to these activities, several studies have demonstrated the benefits of PPAR modulation by BEZ for different pathologies with neurodegeneration and characterized by mitochondrial dysfunction and oxidative stress, such as inherited metabolic disorders [20,21,22]. In this regard, a study from our group previously showed that BEZ prevented changes in mitochondrial function, antioxidant system and neural injury caused by sulfite infusion into the striatum of rats [12].

In this study, we hypothesized that sulfite accumulation in ISOD and MoCD patients causes neuroinflammation and oxidative stress, leading to myelin alterations. To evaluate this possibility, the ex vivo effects of sulfite were studied on these pathomechanisms in the striatum of rats. Viability was also evaluated in MO3.13 cells, a hybrid cell line with similar characteristics to primary oligodendrocytes. Moreover, due to the scarcity of treatment options for ISOD and MoCD, we also assessed whether BEZ could ameliorate sulfite-induced toxicity. 

## 2. Materials and Methods

### 2.1. Reagents

All chemical supplies were from Sigma-Aldrich (St. Louis, MO, USA), unless otherwise stated. Sulfite solution (pH = 7.4) was prepared immediately before each experiment. Corn oil was used as the vehicle for BEZ.

### 2.2. Animals

We used twenty-three-day-old and thirty-day-old Wistar rats, obtained from the Central Animal House of the Department of Biochemistry, ICBS, Universidade Federal do Rio Grande do Sul (UFRGS), RS, Brazil. The animals were maintained under standard conditions. The experimental protocol was approved by the Ethical Committee for Animal Research of UFRGS and followed the National Animal Rights Regulation (Law 11.794/2008) and the National Institutes of Health Guide for the Care and Use of Laboratory Animals (NIH publication 85–23, revised 1996). All efforts were undertaken to minimize the number of animals.

### 2.3. Sulfite Administration

Thirty-day-old male Wistar rats received 2 μmol sulfite (2 μL of 1 M solution) or NaCl at the same dose (control group) bilaterally into each striatum as described [12]. The coordinates used for injection were determined according to Paxinos and Watson (2006) [23]. The animals were euthanized 7 days after sulfite or NaCl injection.

### 2.4. Bezafibrate (BEZ) Treatment

*Post-treatment with BEZ:* We organized the rats as follows: vehicle + NaCl (control), vehicle + sulfite (sulfite), BEZ (30 mg/kg/day) + sulfite (sulfite + BEZ30), and BEZ (100 mg/kg/day) + sulfite (sulfite + BEZ100). The administration of sulfite and BEZ was carried out as previously described [12]. At 37 days of age, the rats were euthanized and the striata were used for immunofluorescence analysis and gene expression evaluation.

*Pre-treatment with BEZ:* In further experiments, a pre-treatment with BEZ was performed as previously described [12,24]. At the age of 37 days, animals were euthanized and striata were dissected.

### 2.5. Immunofluorescence Analysis

Striatum slices (30 μm) prepared in a Leica 1000S vibratome (Nussloch, Germany) were incubated with the antibodies rabbit anti-myelin basic protein (MBP; 1:400, Abcam, Cambridge, UK), rabbit anti-ionized calcium-binding adapter molecule 1 (Iba1; 1:1000, Wako Pure Chemical Industries, Japan), and rabbit anti-neuroglycan-2 (NG2; 1:200, Abcam, Cambridge, UK) as described [12]. After mounting the slices in Fluoroshield™ (Sigma-Aldrich, MO, USA), pictures were taken using a FV300 Olympus confocal microscope (Tokyo, Japan). For each animal and staining procedure, 3 sections were stained. In a separate analysis, three brain slices were directly stained at room temperature with 300 μL of green Fluoromyelin^TM^ (1:300 from the stock solution) (Thermo Fisher Scientific, Waltham, MA, USA) for 30 min. Sections were rinsed, mounted in Fluoroshield™ (Sigma-Aldrich, MO, USA) and imaged using the same microscope. Quantification of fluorescence was performed as described [25,26].

### 2.6. Quantitative RT-PCR 

mRNAs were quantitatively determined via real-time PCR using TaqMan gene expression assays (Applied Biosystems, Thermo Fisher Scientific, USA), as referred for each gene in Appendix A. β-actin was used as a housekeeping gene for the normalization of gene expression [27,28].

### 2.7. MO3.13 Cell Culture 

MO3.13 cells (RRID: CVCL_D357; kindly provided by Prof. Daniel Martins de Souza, UNICAMP, Brazil) consist of an immortal oligodendrocytic human–human hybrid cell line [29]. MO3.13 cells were cultured in a growth medium (GM; DMEM, supplemented with 15 mM HEPES, 14.3 mM sodium bicarbonate, 100 IU/mL penicillin and 100 μg/mL streptomycin), and 10% *v*/*v* fetal bovine serum (FBS) at 37 °C under 5% CO_2_ atmosphere until reaching confluence. The number of passages was maintained between 10 and 16. For the experiments, 3 × 10^4^ cells/cm^2^ were seeded.

### 2.8. Exposure of MO3.13 Cells to Sulfite and BEZ and Viability

On the day of the experiment, sulfite was dissolved in PBS, pH 7.4, and the solution was sterilized via filtration (0.22 µm membrane). At 80% of confluence, MO3.13 cells were exposed to sulfite (1–1000 μM) prepared in GM with 5% FBS for 24–48 h (37 °C; 95% air/5% CO_2_ incubator). Further experiments consisted of the post- or pre-treatment of cells with BEZ (200 or 1000 nM) for 6 h. Cells were incubated for 24 h with sulfite (500 μM) before or after the addition of BEZ. Resazurin reduction assay was carried out to determine viability, as previously reported [30].

### 2.9. Statistical Analysis

Assays were performed in duplicate or triplicate and the mean ± standard deviation is shown in the figures. Data were analyzed using the SPSS software (New York, NY, USA). One-way ANOVA followed by the post hoc Duncan multiple range test was used when F was significant. Differences were significant when *p* < 0.05. 

## 3. Results

### 3.1. BEZ Post-Treatment Prevents Sulfite-Induced Changes on Myelin Structure in the Striatum and Reduced the Viability of MO3.13 Cells

First, we assessed the effects of sulfite and BEZ administration on myelin structure in the striatum of rats. Sulfite administration significantly decreased the fluorescence of MBP, an important marker of mature oligodendrocytes [31], and caused fragmentation of striatal axonal bundles (Figure 1A). Post-treatment with BEZ attenuated these effects with the 100 mg/kg/day dose, but not the 30 mg/kg/day dose (Figure 1A). Consistent with these results, staining with the Fluoromyelin^TM^ probe also demonstrated that sulfite causes disorganization and fragmentation of axonal bundles, which was prevented by 100 mg/kg/day BEZ (Appendix A).

Remyelination may be induced by different insults that cause white matter alterations [32], so we examined the staining of NG2, a marker of oligodendrocyte progenitor cells (OPCs). Sulfite injection increased NG2 staining in the striatum, which was prevented by post-treatment with 30 mg/kg/day but not with 100 mg/kg/day of BEZ (Figure 1B).

The effects of sulfite and BEZ on the viability of MO3.13 and on the in vitro model used for the study of oligodendrocytes, were evaluated next. Sulfite (500 and 1000 μM) significantly decreased cell viability after 24 or 48 h incubation (Figure 2A,B), and this effect was prevented by BEZ (1 μM) after incubation for 24 h (Figure 2C).

### 3.2. BEZ Post-Treatment Prevents Sulfite-Induced Neuroinflammation and Oxidative Stress in the Striatum

Microglial activation with consequent elevation of pro-inflammatory factors has been depicted in some demyelinating conditions [33,34]. Thus, we examined the effects of sulfite exposure and BEZ administration on Iba1 staining and the expression of genes involved in inflammation and redox status. Sulfite markedly increased Iba1 staining, suggesting microglial activation (Figure 3). Post-treatment with 30 mg/kg/day BEZ attenuated and 100 mg/kg/day BEZ totally prevented Iba1 change (Figure 3).

Next, we evaluated the effects of sulfite and BEZ on the striatal inflammatory response. Sulfite significantly increased the expression of IL-6 (Figure 4A) COX-2 (Figure 4H) and IL-1β (Figure 4B), while it reduced IL-10 (Figure 4C). BEZ post-treatment (30 and 100 mg/kg/day) attenuated the effect on IL-6 (Figure 4A) and further increased the mRNA levels of IL-1β (Figure 4B). Additionally, BEZ prevented the expression of IL-10 with 30 mg/kg/day, but increased it with 100 mg/kg/day (Figure 4C). BEZ did not alter the sulfite effect on COX-2 (Figure 4H). Sulfite per se did not alter TNF-α (Figure 4D) and NFκB p65 compared to the control group (Figure 4G). However, in the striatum of rats administered with sulfite and BEZ (30 and 100 mg/kg/day), TNFα expression was increased and NFκB p65 was decreased. IL1R1 and TNFR1 mRNA levels did not change under any condition (Figure 4E,F).

Of genes implicated in oxidative stress induction, sulfite exposure increased iNOS (Figure 5A) and reduced HO-1 mRNA levels (Figure 5C), but did not change Nrf2, SOD1 and SOD2 expression (Figure 5B,D,E). Post-treatment with 100 mg/kg/day BEZ prevented the effects on iNOS and HO-1 (Figure 5A,C).

### 3.3. BEZ Pre-Treatment Prevents Sulfite-Induced Alterations on Myelin Structure and Neuroinflammation in the Striatum, and Reduced Viability of MO3.13 Cells

Finally, we studied the effects of pre-treatment with BEZ (rat model: 30 or 100 mg/kg/day; and MO3.13 cell model: 200 or 1000 nM) on the toxic effects caused by sulfite on rat striatum and MO3.13 cells. While 30 mg/kg/day BEZ ameliorated the sulfite-induced decrease in MBP staining (Figure 6A), 30 and 100 mg/kg/day BEZ attenuated the effect on Iba1 staining (Figure 6C). In MO3.13 cells, the sulfite-induced decrease in viability at 24 h was prevented by BEZ (200 and 1000 nM) (Figure 6B).

## 4. Discussion

ISOD and MoCD are life-threatening diseases that causes severe neurological deterioration due to the accumulation of sulfite in the brain. White matter changes are commonly observed but their pathophysiology is still unknown. Moreover, therapy is not effective and a high mortality rate can be observed [1,5]. To investigate the pathophysiology of brain abnormalities with a focus on white matter alterations, we examined the effect of sulfite exposure on myelin structure, neuroinflammation, and oxidative stress in a rat model with a direct striatal injection of this sulfur metabolite. We also investigated the neuroprotective effects of the pan-PPAR agonist BEZ treatment before and after sulfite exposure. 

Sulfite indeed led to myelin loss in the striatum as evidenced by decreased staining of myelinated axons observed with the use of MBP and Fluoromyelin^TM^ fluorescence. Notably, sulfite reduced the viability of MO3.13 cells, indicating that it is toxic to oligodendrocytes. In this regard, it should be highlighted that these cells express different proteins that are found in immature oligodendrocytes, including MBP [35], rendering them as a useful model to study oligodendrocyte injury.

Sulfite also increased NG2 staining, suggesting an increased number of OPCs, the predominant cell type involved in remyelination. Mounting evidence indicates that OPCs are able to respond to different insults by proliferating and differentiating into myelinating oligodendrocytes [36], which is consistent with the findings of oligodendrocyte and myelin changes in a number of neurodegenerative disorders [37]. 

Given that neuroinflammation with microglial activation is commonly involved in disorders characterized by white matter alteration [38,39], we decided to evaluate whether sulfite could cause a neuroinflammatory response. Sulfite exposure induced neuroinflammation in the striatum as evidenced by the increased staining of Iba1 and the elevated expressions of IL-1β, IL-6 and COX-2, along with a reduction of IL-10. Increased Iba1 also suggests a compensatory mechanism aiming for remyelination since this process requires microglia activation for the clearance of myelin debris and OPC activation [40]. In addition, the augmented synthesis and secretion of factors by activated microglia and astrocytes, such as IL-1β, has been shown to result in increased migration and differentiation of activated OPCs [41]. This is in line with our data on NG2 staining. In addition, we have previously demonstrated that sulfite increases GFAP and S100B, indicating that high IL-1β mRNA levels may be derived from astrocytes [12].

Sulfite-induced alterations on myelin and inflammatory markers in rat striatum and the decreased viability of MO3.13 cells were mitigated by treatment with BEZ following exposure. The reduction in NFκB and an elevation in TNF-α and IL-1β were observed only in BEZ-treated groups, implying that this molecule may induce both pro-inflammatory and anti-inflammatory effects. Interestingly, only the lowest dose of BEZ mitigated the effect of sulfite on NG2 staining, implying that some effects not yet elucidated caused by 100 mg/kg/day BEZ may hamper its protective potential on this parameter. In line with this, some data revealed toxic effects of BEZ in different cells and tissues, such as immune cells and the liver [42,43]. Therefore, since previous data show that BEZ induced anti-inflammatory effects in different models of inherited metabolic disorders [22], additional studies on BEZ dosing levels and timing are warranted to optimize the desired response in inflammation.

We have previously shown that in vivo and in vitro exposure of rat brain to sulfite induces oxidative stress by increasing ROS levels, eliciting lipid peroxidation, and impairing antioxidant defenses [8,12,13,44]. Since neuroinflammation is often correlated with elevated ROS [45,46], we characterized the expression of genes related to antioxidant defenses and signaling following sulfite exposure. Sulfite administration markedly increased iNOS expression, suggesting oxidative stress induction and reinforcing the likelihood that this metabolite induces neuroinflammation. The accumulation of nitric oxide along with ROS, as previously demonstrated for sulfite exposure [44], results in the formation of different reactive nitrogen species capable of protein nitration and eventually cell death [47]. Furthermore, in conjunction with the alterations in the cytokines and Iba1, we speculate that sulfite induces microglial polarization to M1, which is associated with a pro-inflammatory phenotype [48]. In addition, the reduction in HO-1 expression that we found is consistent with the literature showing that sulfite also reduces the protein content of this enzyme [13]. Surprisingly, the expression of Nrf2, a well-known regulator of HO-1, was not modified. We suggest that Keap-1, the negative regulator of Nrf2 [49], is degraded so HO-1 may be upregulated by sulfite without changes in Nrf2 expression. Moreover, we previously demonstrated that SOD1 protein content was increased by sulfite [13] whereas the gene expression was not altered in the present work, suggesting a modulation at the translation level. It should be further noted here that oligodendrocytes have high iron content [50] and that sulfite generates high levels of free radicals in the presence of ferric ions [51], implying that oxidative stress likely mediates myelin disruption. Post-treatment with BEZ prevented the alterations in iNOS and HO-1 expression, which is in line with the previously protective effects of this drug observed on the inflammatory response markers. It is also consistent with reports suggesting that sulfite induces oxidative stress in the brain [8,12,13,44].

Nevertheless, there are limitations in our study. The mechanisms involved in sulfite-induced myelin damage were not elucidated. Although the injury to MO3.13 cells suggests that sulfite elicits toxicity directly to oligodendrocyte cells, we cannot rule out that neuronal damage caused by this metabolite may cause a secondary effect on myelin. In this sense, neuronal loss was caused by sulfite in the striatum of rats, which was prevented by BEZ [12]. Furthermore, it is hard to extrapolate the doses of BEZ used in the present study to human pathologies. Although the doses of BEZ used here are in accordance with previous works performed in rodent models, lower doses have been utilized in patients with other inherited metabolic disorders (up to 800 mg/day), such as fatty acid oxidation defects [22]. However, it should be noted that longer therapeutical regimens are often evaluated in patients, which enables the use of lower doses with efficacy. It should be also considered that some controversial findings on the benefits of BEZ were found. In this regard, leriglitazone, a specific PPARγ agonist, was recently shown to elicit remarkable effects in several models of X-linked adrenoleukodystrophy, including remyelination and neuroinflammation reduction [34], suggesting that target exposure and specificity might be important characteristics for the efficacy of PPAR agonists.

In summary, our data show that sulfite exposure impairs myelin structure in the striatum, possibly mediated by neuroinflammation associated with oxidative stress. Although post- and pre-treatment with BEZ mitigated most of the alterations, secondary effects on cytokine expression were induced by this drug, suggesting that additional dose levels and timing of BEZ and other more specific PPAR agonists should be evaluated. Overall, our findings suggest that the use of PPAR agonists to prevent neuroinflammation and myelin disruption is a promising strategy for the pharmacological management of ISOD and MoCD. 

## Figures and Tables

**Figure 1 cells-12-01557-f001:**
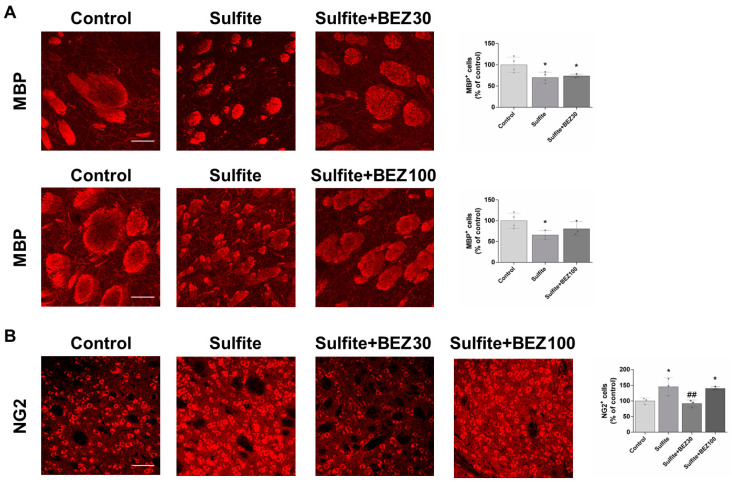
Bezafibrate (BEZ) post-treatment mitigates sulfite-induced decrease in MBP and increase in NG2 staining in rat striatum 7 days after sulfite injection (2 μmol). Animals were post-treated with BEZ [30 (**A**) or 100 (**B**) mg/kg/day] for 7 days after sulfite infusion. *n* = 3–4. * *p* < 0.05, compared to rats receiving NaCl (2 μmol) (control group); ## *p* < 0.01, compared to rats receiving sulfite (sulfite group) (ANOVA followed by Duncan multiple range test).

**Figure 2 cells-12-01557-f002:**
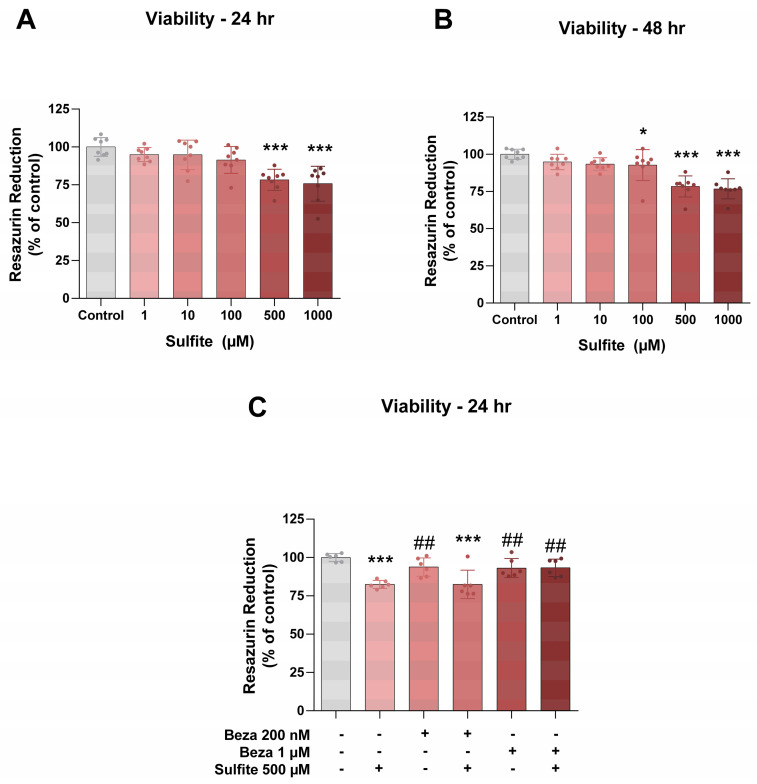
Bezafibrate (BEZ) post-treatment mitigates sulfite-induced decrease in the viability of MO3.13 cells. Cells were incubated with sulfite (500 μM) for 24 or 48 h (**A**,**B**). In further experiments, cells were post-treated with BEZ (200 or 1000 nM) for 6 h (**C**). *n* = 6–8. * *p* < 0.05, *** *p* < 0.001, compared to control group; ## *p* < 0.01, compared to sulfite group (ANOVA followed by Duncan multiple range test).

**Figure 3 cells-12-01557-f003:**
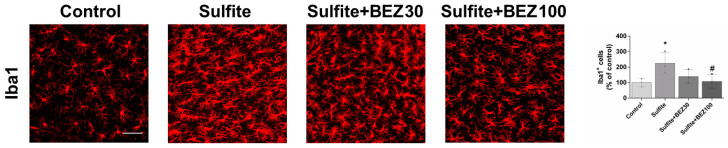
Bezafibrate (BEZ) post-treatment mitigates sulfite-induced increase in Iba1 staining in rat striatum 7 days after sulfite injection (2 μmol). Animals were post-treated with BEZ (30 or 100 mg/kg/day) for 7 days after sulfite infusion. *n* = 3. * *p* < 0.05, compared to rats receiving NaCl (2 μmol) (control group); # *p* < 0.05, compared to rats receiving sulfite (sulfite group) (ANOVA followed by Duncan multiple range test).

**Figure 4 cells-12-01557-f004:**
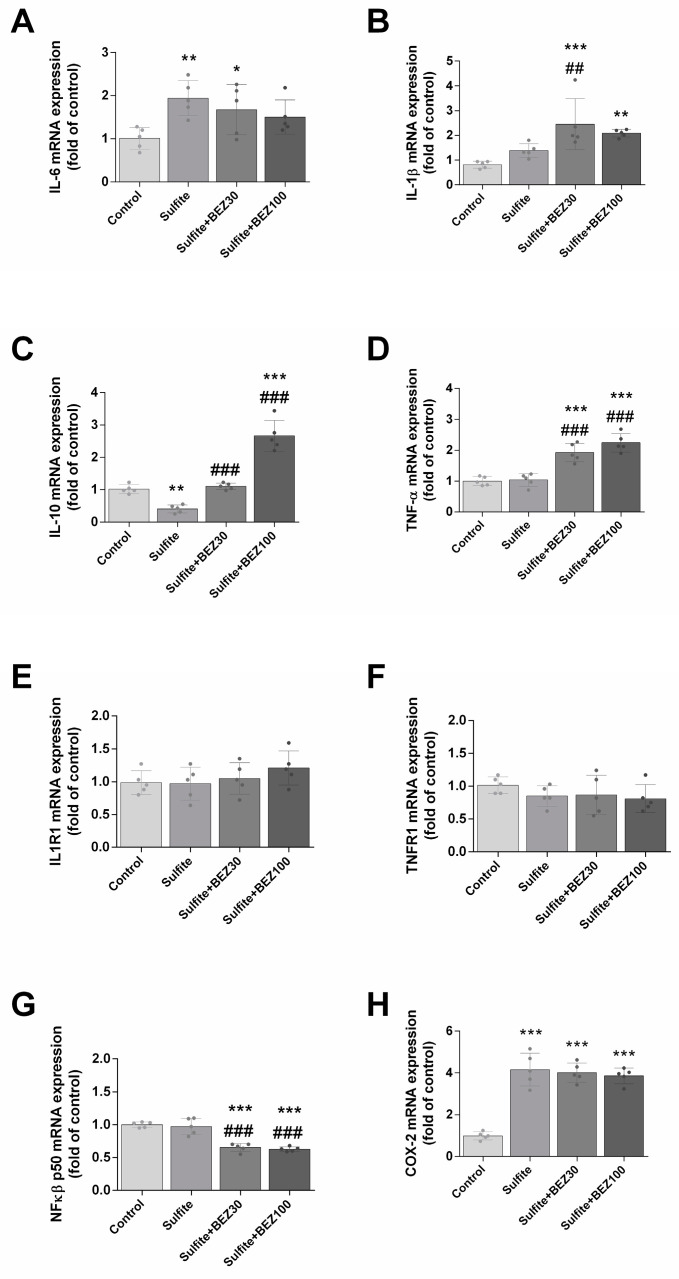
Alterations on the expression of IL-6 (**A**), IL-1β (**B**), IL-10 (**C**), TNF-α (**D**), ILR1 (**E**), TNFR1 (**F**), NFκB (**G**) and COX-2 (**H**) elicited by bezafibrate (BEZ) post-treatment and sulfite in rat striatum 7 days after sulfite injection (2 μmol). Animals were post-treated with BEZ (30 or 100 mg/kg/day) for 7 days after sulfite infusion. *n* = 5. * *p* < 0.05, ** *p* < 0.01, *** *p* < 0.001, compared to rats receiving NaCl (2 μmol) (control group); ## *p* < 0.01, ### *p* < 0.001, compared to rats receiving sulfite (sulfite group) (ANOVA followed by Duncan multiple range test).

**Figure 5 cells-12-01557-f005:**
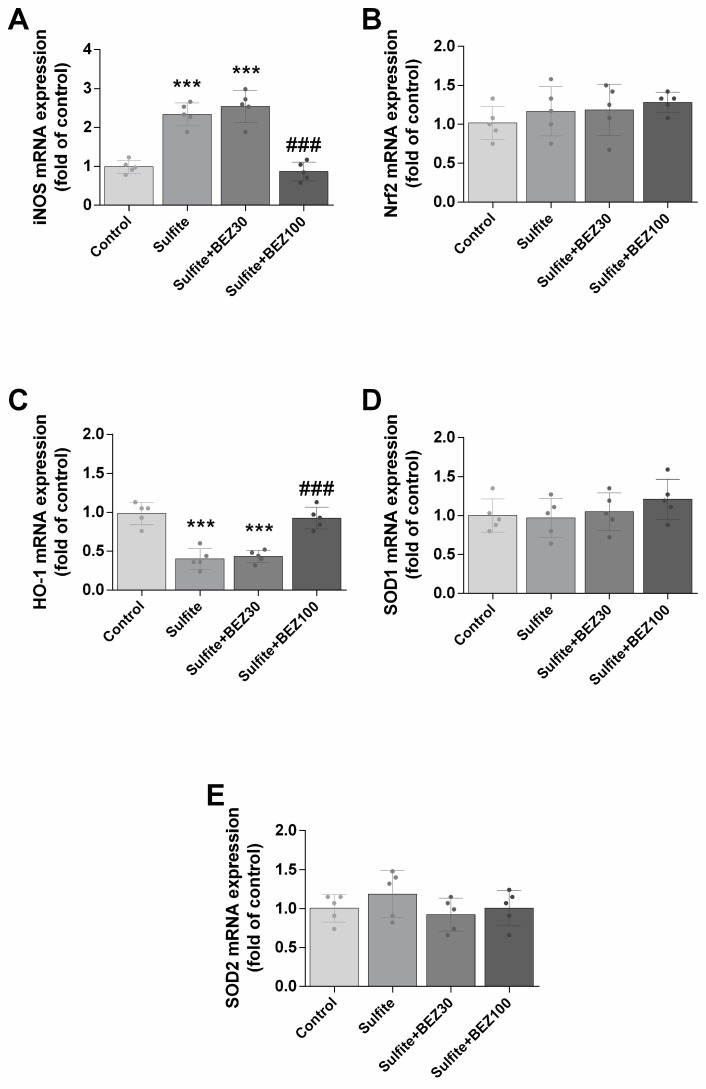
Alterations on the expression of iNOS (**A**), Nrf2 (**B**), HO-1 (**C**), SOD1 (**D**) and SOD2 (**E**) elicited by bezafibrate (BEZ) post-treatment and sulfite in rat striatum 7 days after sulfite injection (2 μmol). Animals were post-treated with BEZ (30 or 100 mg/kg/day) for 7 days after sulfite infusion. *n* = 5. *** *p* < 0.001, compared to rats receiving NaCl (2 μmol) (control group); ### *p* < 0.001, compared to rats receiving sulfite (sulfite group) (ANOVA followed by Duncan multiple range test).

**Figure 6 cells-12-01557-f006:**
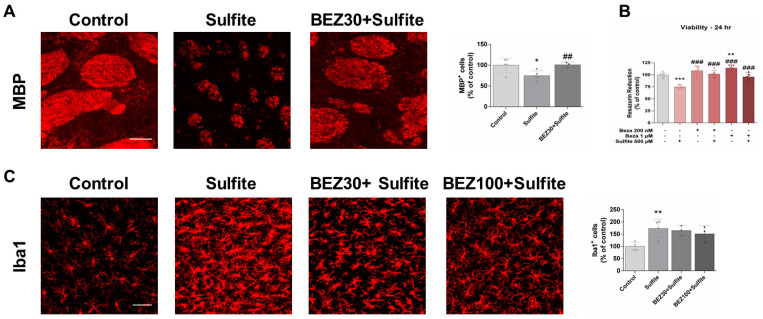
Bezafibrate (BEZ) pre-treatment mitigates sulfite-induced alterations on MBP and Iba1 staining in rat striatum, and reduction in the viability of MO3.13 cells. Animals were pre-treated with BEZ (30 or 100 mg/kg/day) for 7 days after sulfite infusion (**A**,**C**). In MO3.13 cells, pre-treatment with BEZ (200 or 1000 nM) was for 6 h (**B**). *n* = 3–6. * *p* < 0.05, ** *p* < 0.01, *** *p* < 0.001, compared to rats receiving NaCl (2 μmol) (control group); ## *p* < 0.01, ### *p* < 0.001, compared to rats receiving sulfite (sulfite group) (ANOVA followed by Duncan multiple range test).

## Data Availability

The data presented in this study are available on request from the corresponding author. The data are not publicly available due to private reasons.

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
