# Peer review of "Myelin Disruption, Neuroinflammation, and Oxidative Stress Induced by Sulfite in the Striatum of Rats Are Mitigated by the pan-PPAR agonist Bezafibrate"

_cells, 2023, doi:10.3390/cells12121557_

Round 1

Reviewer 1 Report

I regret to say that article titled “Myelin disruption, neuroinflammation, and oxidative stress induced by sulfite in the striatum of rats are mitigated by the pan-PPAR agonist bezafibrate" by Glänzel, et al. isn’t suitable for publication in cells.
 Several factors are important for my decision, including:

 The potential scientific impact;
The author did not elaborate on why MO3.13 cells were used as bezafibrate to protect the background of Myelin disruption, neuroinflammation, and oxidative stress and the purpose of this study.
There are many conflicting phenomena in the results, such as

1.     In Fig1 is not clear and why the BEZ100 group in Figure 1B increases the performance of NG2. It is also suggested to integrate Fig.1A into one result.

2.     In Fig. 3, why using the expression of Iba1 to explain the phenomenon of inflammation and oxidative stress, and is not clear? The preface pointed out that BEZ has the ability of anti-inflammation and anti-oxidation, but this result indicated that BEZ induced the mRNA expression of IL-1β, IL-1R1, and TNF-a. It is suggested that the authors should perform protein expression analysis. 

Reviewer 2 Report

Dear colleagues!
The paper by Glanzel et al. addresses an important issue in modern medicine and provides a descriptive proof of bezafibrate potential for treatment of sulfite-induced striatum damage. The paper is concise and well-written to describe the performed study.

Nevertheless, the Reader may raise the following questions on strength of the work (while Authors do describe its limitations in Discussion section):

1)    Histological images lack nuclei staining which deviates from typically spread technique and results in

2)    question whether the sting structure counts were normalized to cell counts by nuclei as far as % of control was used and in Figures 1A and 3 structure size variation is quite vivid

3)    The paper lacks a good schematic to connect the findings between each other in Discussion section and certain interconnections are definitely of speculative nature and do not rise directly from descriptive analysis made in the work.

4)    Majority of targets identified in the study are assayed by RT-PCR and are not validated by WB or ELISA and thus require additional experimental procedures to improve the work’s strength.

Regards, Reviewer

Round 2

Reviewer 2 Report

Dear colleagues!
I appreciate the revisions made.

Regards, Reviewer

Author Response

Dear reviewer,
We appreciate the suggestions and comments. Best regards!